# Grassland restoration in typical wind-eroded regions effectively increase soil organic carbon

**Bin Xia**[1,2], **Wei Xu**[1]*

1 School of Humanities and Social Sciences, Jiangsu University of Science and Technology, Zhenjiang, China, 2 Institute of Soi and Water Conservation, Chinese Academy of Sciences and Vinistry of Water Resources, Yangling, Shaanxi, China

* xw20217@just.edu.cn

## Abstract

Soil organic and inorganic carbon (SOC and SIC, respectively) are the two most important carbon pools in the terrestrial carbon cycle, yet their responses to land use change in typical wind-eroded regions remain poorly understood. This study analyzed the carbon change patterns of four land use types Yanchi County, including the seasonal dynamics and driving factors of SOC, SIC, and total carbon storage under wind erosion background. According to filed measurement, the SOC and SIC contents in cropland were $3.0\,g\,kg^{-1}$ and 12%, respectively. Compared with cropland, grassland restoration markedly increased SOC to $4.4\,g\,kg^{-1}$ but reduced SIC to 2.7%, primarily due to enhanced organic matter inputs and the suppression of wind erosion. In contrast, shrubland restoration resulted in lower SOC ($\sim 2.4\,g\,kg^{-1}$) and SIC ($\sim 2.5\%$) contents, likely because the slow decomposition of recalcitrant litter and coarse root biomass limited carbon turnover. Both SOC and SIC exhibited distinct vertical distribution patterns with depth, with SOC mainly concentrated in the 0–1 cm layer and SIC in the 1–5 cm layer. These contrasting profiles can largely be attributed to their dominant controlling factors: SOC was primarily regulated by vegetation cover, whereas SIC was strongly influenced by soil pH. Nevertheless, both carbon pools were sensitive to variations in wind erosion intensity and soil texture. These findings highlight distinct control processes over SOC and SIC, as well as underscore the surface soil (0–5 cm) as a critical interface mediating vegetation, erosion, and soil properties.

## Introduction

The soil carbon pool is one of the most important components of terrestrial carbon cycle, and its dynamic changes directly regulate atmospheric $CO_2$ and feedbacks to climate change [1,2,3,4]. Soil organic carbon (SOC) and soil inorganic carbon (SIC) are the two dominant pools of soil carbon. The global SOC stocks to 2 m of soil depth are estimated at approximately 2400 Pg [5]. In arid and semi-arid regions, inorganic

**Data availability statement:** The SOC, SIC data of the manuscript are available from the Figshare database after acceptance. And DOI is https://doi.org/10.6084/m9.figshare.30597233.v1.

**Funding:** This research was funded by the Research Start-up Grant for Invited Teachers of Jiangsu University of Science and Technology Award Recipient Initials: W.X. Grant Number: No. 1192932203 & 2195082503 Funder: Jiangsu University of Science and Technology Funder Website: https://www.just.edu.cn/ Role of Funders: The funders played a role in the conceptualization, validation, investigation, and decision to publish of the manuscript.

**Competing interests:** The authors have declared that no competing interests exist.

carbon stored as pedogenic carbonates can account for a substantial proportion of total soil carbon. soils store $2305 \pm 636$ Pg of carbon as SIC within the upper 2 m, a magnitude comparable to or even exceeding global SOC stocks. Importantly, this pool is not static. Projections suggest that global SIC stocks (0–30 cm) will reduce by up to 23 Pg of carbon [6,7]. SOC is primarily controlled by vegetation inputs, microbial decomposition, and soil physicochemical properties [8,9,10]. And SIC is affected by mineral weathering and soil pH [11,12]. The storage and spatial distribution of these two pools are not only regulated by natural factors such as climate, geomorphology, and maternity, but also by artificial activities including agricultural management, and ecological restoration [13,14,15]. Understanding the spatiotemporal variability and controlling factors of SOC and SIC is a frontier issue in global climate change and earth science research.

In arid and semi-arid regions, wind erosion is a major process driving soil degradation and altering carbon cycling. Through detachment, transport, and deposition, wind erosion preferentially removes nutrient rich surface soils. Those processes lead to the redistribution of SOC and SIC, changing terrestrial carbon balances [13,16,17]. The aeolian processes selectively transport small and carbon-rich particles, causing SOC loss, particularly at the 1 cm depth severely affected by wind erosion [18,19]. The annual surface SIC loss in global arid regions has been confirmed to be 11.33 g C m$^{-2}$ [20], and SOC is also as high as 0.22–0.53 Pg [8,21]. Furthermore, small variations in the uppermost millimeters to centimeters of soil aggregation, crusting, and moisture can substantially alter erosion thresholds, thereby changing the redistribution process [22]. However, this effect is highly heterogeneous across different land use types and seasons. Therefore, climate warming and changes in land use will further amplify such losses. [23], focusing on the dynamics of surface carbon pools has become crucial for assessing the future direction of carbon balance in wind-eroded regions.

The Ningxia Hui Autonomous Region of northern China, particularly Yanchi County, lies in the transitional zone between the Loess Plateau and the Inner Mongolian Plateau. This region is highly sensitive to wind erosion, wind erosion modulus in this area exceeds 3000 t km$^{-2}$ yr$^{-1}$, which belongs to the moderate to severe erosion [24,25]. It has long faced ecological problems such as desertification, vegetation degradation, and soil fertility decline [26,27]. As a key region of the national "Grain-for-Green" program, Yanchi has experienced substantial land use changes. The area of cropland has decreased, and grasslands and shrublands have gradually increased. While these transformations have reduced land degradation and improved ecological conditions, they have also altered surface roughness, vegetation cover, soil structure, and nutrient distribution, potentially reshaping SOC and SIC dynamics. Previous studies have identified the major environmental drivers of SOC [28,29] and SIC [20,30,31], but considering these two pools separately. Lack of research on wind erosion has limited our understanding of how land use change and wind erosion processes jointly regulate SOC and SIC in erosion-prone ecosystems.

To address these gaps, this study focuses on Yanchi County as a representative wind erosion region. We examined soils under four land use types: cropland (continuously cultivated), grasslands abandoned for 10 and 20 years (naturally restored after

cultivation cessation), and shrubland (established through afforestation on former cropland). Soil samples were collected at two key surface depths (1 cm and 5 cm) across multiple seasons. By integrating measurements of wind regimes (seasonal wind energy index and erosion thickness), plant cover, and soil physicochemical properties (soil water content, bulk density, texture, pH, and compactness), we aimed to systematically reveal the spatiotemporal dynamics of SOC and SIC and their driving mechanisms. Specifically, we addressed research hypotheses: 1) Grassland and shrubland restoration would be the land-use practice for enhancing surface SOC, primarily through increased plant cover and reduced wind erosion. 2) SIC represents a significant and dynamically responding pool in dryland region. 3) The heterogeneity of both SOC and SIC in dryland region is predominantly driven by wind erosion processes by physically redistributing soil particles. This study was designed to quantify the spatiotemporal variations in SOC and SIC across different land use types, soil depths, and seasons, to compare their distribution patterns between the surface soil layer (0–1 cm) most directly affected by wind erosion and the subsurface layer (1–5 cm), and to disentangle the relative contributions of wind erosion versus land use practices in shaping SOC and SIC heterogeneity while elucidating the underlying mechanisms. This study is not only essential for improving the accuracy of regional soil carbon stock assessments but also providing a scientific basis for designing land use and ecological restoration strategies in arid and semi-arid regions.

## Materials and methods

### Study area

The study was conducted in Yanchi County, Ningxia Hui Autonomous Region of northern China, situated in the ecotone between the Loess Plateau and the Inner Mongolian Plateau (37°04′–38°10′ N, 106°30′–107°41′ E) (Fig 1). The county covers an area of approximately 6,778 km$^2$, with elevations ranging from 1,300–1,600 m above sea level. The region has a typical temperate semi-arid continental monsoon climate, characterized by cold, dry winters and hot summers. Mean annual precipitation is approximately 250–350 mm, over 70% of which falls between June and September, typically as short-duration, high-intensity storms. The mean annual temperature is about 8.7 °C, and potential evaporation is around 2,400 mm yr$^{-1}$, resulting in a pronounced moisture deficit. The mean annual wind speed is about 2.7 m s$^{-1}$, with about 24 gale days (days with daily maximum wind speed ≥17.0 m s$^{-1}$ at the 10 m height [32]) and 21 sandstorm days per year, making this one of northern China's representative wind erosion-prone regions. Soils are dominated by gray calcareous soils (sierozems) and wind-deposited sandy soils in the north, transitioning to dark loessial soils in the south. The region is part of the semi-arid grassland desert steppe transition zone, where natural vegetation consists mainly of desert steppe communities dominated by xerophytic shrubs such as *Caragana korshinskii* and *Artemisia ordosica*, along with perennial grasses like *Stipa bungeana* and *Leymus secalinus*. Decades of overgrazing and cultivation have led to severe land degradation, though recent ecological restoration efforts, such as the "Grain-for-Green" program and grazing exclusion, have facilitated vegetation recovery and reduced wind erosion intensity.

### Experimental design and soil sampling

To investigate the effects of land use type and wind erosion processes on soil carbon pools, a field experiment was established in 2021. Four typical land use types were selected: 1) cropland continuously cultivated under traditional farming, and dominated by *Fagopyrum esculentum Moench*. 2) grassland10 abandoned from cropland in 2011 (10 years prior to sampling), and dominated by *Artemisia scoparia*, *Leymus secalinus*, *Heteropappus altaicus*. 3) grassland20 abandoned from cropland in 2001 (20 years prior to sampling) and dominated by *Agropyron cristatum*, *Leymus secalinus*, *Heteropappus altaicus*. Both grassland10 and grassland20 have naturally recovered without any human interference (e.g., grazing, mowing, or reseeding) since abandonment. Before abandonment, they were managed identically to the cropland. 4) shrubland converted from cropland in 2011 (10 years prior to sampling) by planting shrubs (dominated by *Salix pasmmophara*, *Hedysarum scoparium*), it has been protected from grazing and other disturbances since establishment. These land use types can represent the main plant ecological conditions in the area under the ecological restoration project.

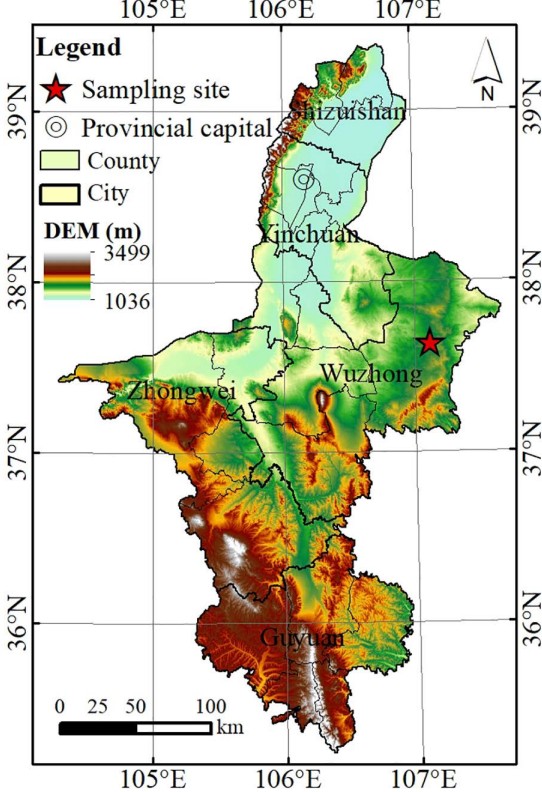

**Fig 1. Location of the study area.**

Soil samples were collected at two depths (0–1 cm and 0–5 cm) with a weight greater than 5 g. According to the measurement of erosion thickness (Table 1), the 0–1 cm layer was the most vulnerable to wind detachment at the annual scale. The 0–5 cm layer encompassed the main active layer, and its properties largely determined the surface resistance to erosion [33]. To capture seasonal dynamics, field sampling was conducted four times per year (spring, summer, autumn, and winter) during 2021 and 2022. Within each plot (approximately 10 m × 10 m), 5 replicate sampling points were selected using an S-shaped random sampling strategy. The approximate distance between successive sampling points was 3 m. Soils were composited for laboratory analysis. The field experiment was initiated in early 2021, and soil and environmental monitoring was carried out for two consecutive years (2021–2022). Seasonal sampling was conducted from March 10–15 (spring), June 10–15 (summer), September 10–15 (autumn), and December 10–15 (winter), representing the main wind erosion and vegetation growth periods in study region.

### Wind and erosion measurements

To characterize wind conditions, a portable micro-meteorological station was installed in each site to record wind speed and wind direction at 10-minute intervals. Based on these observations, wind erosion index (WEI) was calculated using the modifiedwind energy formula [34]:

$$WEI = \frac{1}{100} \sum_{i=1}^{n} \overline{V}^3 \left( \frac{PET_i - P_i}{PET_i} \right) d$$

(1)

**Table 1. Differences in soil properties among land use types and seasons in a typical wind erosion region of sampling sites (mean±sd).**

| Land use | Season | BD (g cm⁻³) | pH | WEI | Plant cover (%) | Erosion thickness (cm yr⁻¹) | WC (%) | Compactness (kPa) | Clay (%) | Silt (%) | Sand (%) |
|---|---|---|---|---|---|---|---|---|---|---|---|
| cropland | spring | 1.4±0.3B | 8.1±0.4A | 34.24±24.40Aa | 5.15±5.23Db | 0.27±0.10Aa | 1.11±0.31Ab | 21.4±11.1Dc | 6.3±1.5Ac | 10.1±2.1Ac | 83.6±3.5Ca |
|  | summer |  |  | 23.42±12.43Ab | 32.04±9.65Ca | 0.12±0.03Ac | 3.79±3.65Ab | 12.3±6.4 Cd | 7.6±1.9Ab | 11.1±2.4Abc | 81.3±4.2Bab |
|  | autumn |  |  | 23.14±23.22Ab | 0±0Dc | 0.16±0.05Abc | 0.61±0.19Ac | 75.9±1.6Ca | 10.0±2.5Ca | 14.9±3.5Aa | 75.1±5.9Bc |
|  | winter |  |  | 26.91±53.60Ba | 0±0Dc | 0.29±0.14Aa | 6.74±1.53Ba | 59.3±10.0Db | 8.2±1.5Ab | 12.5±1.8Ab | 79.4±3.2Bb |
| grassland10 | spring | 1.64±0.13A | 7.9±0.3AB | 33.94±19.80Aa | 23.15±6.53Cb | 0.13±0.04Ba | 0.29±0.16Bb | 147.2±25.9Aa | 2.1±0.5Ba | 3.9±0.6Ba | 94.0±1.1Bb |
|  | summer |  |  | 25.21±15.26Aa | 34.49±7.64Ca | 0.09±0.10Ba | 1.35±0.91Ac | 60.5±4.9Bd | 1.5±0.4Bb | 2.7±0.5Cb | 95.8±0.9Aa |
|  | autumn |  |  | 26.51±20.18Aa | 28.82±8.84Cb | 0.10±0.01Ba | 0.61±0.19Ad | 135.9±15.9Ab | 1.8±0.6Cab | 3.3±0.8Bab | 94.9±1.4Aab |
|  | winter |  |  | 49.21±72.72Aa | 21.73±7.72Cb | 0.14±0.05Ba | 6.34±0.30Ba | 114.1±2.6Ac | 2.0±0.5Ba | 3.6±0.6Ba | 94.5±1.1Ab |
| grassland20 | spring | 1.56±0.08A | 7.9±0.2AB | 21.55±10.46Ba | 68.13±6.05Aab | 0.02±0.01Ca | 0.25±0.26Bc | 59.7±7.1Cd | 2.4±0.8Ba | 4.4±1.1Ba | 93.2±1.8Bb |
|  | summer |  |  | 14.92±9.36Bb | 77.69±8.33Aa | 0.01±0.00Ca | 1.05±0.76Ab | 79.3±3.2Ab | 1.6±0.6Bb | 3.8±1.0Ba | 94.6±1.6Aa |
|  | autumn |  |  | 14.52±10.76Bb | 66.11±6.32Ab | 0.02±0.00Ca | 1.36±1.21Ab | 91.9±7.7Ba | 1.8±0.7Cab | 3.8±1.3Ba | 94.4±2.0Aab |
|  | winter |  |  | 19.52±23.04Ba | 62.06±5.58Ab | 0.03±0.01Ca | 5.34±0.38Ca | 75.2±3.8Cc | 2.1±0.6Ba | 4.1±1.0Ba | 93.8±1.6Aab |
| shrubland | spring | 1.6±0.04A | 7.7±0.2B | 18.88±9.42Ba | 42.13±7.32Bbc | 0.00±0.00Da | 0.21±0.15Bc | 67.8±6.9Bc | 1.4±0.4Ca | 3.4±0.8Ca | 95.3±1.2Aa |
|  | summer |  |  | 6.00±7.93Cc | 54.15±14.25Ba | 0.00±0.00Da | 0.64±0.23Bb | 82.2±7.3Ab | 1.2±0.7Ba | 3.2±1.6Ba | 95.6±2.3Aa |
|  | autumn |  |  | 6.09±6.07Cc | 47.72±7.90Bab | 0.00±0.00Da | 1.31±1.23Ab | 90.2±3.2Ba | 1.8±0.8Ca | 4.2±1.4Ba | 94.1±2.3Aa |
|  | winter |  |  | 11.75±9.96Bb | 37.64±9.11Bc | 0.00±0.00Da | 8.11±1.39Aa | 80.0±0.7Bb | 1.6±0.5Ca | 3.8±1.0Ba | 94.7±1.4Aa |

Note: BD, bulk density (g cm⁻³); WEI, wind erosion index; WC: soil water content (%). Different uppercase letters (A, B, C) within the same column indicate significant differences among land use types within the same seasons, while different lowercase letters (a, b, c) indicate significant differences among seasons within the same land use type

where $\overline{V}$ is the monthly average wind speed at 2 m height (m s$^{-1}$), $PET_i$ is monthly potential evaporation (mm), $P_i$ is the monthly average precipitation (mm), and $d$ is number of days.

Soil erosion intensity was monitored using 9 erosion needles, which were arranged in an S-shaped layout within each plot. In order to estimate the results of erosion or sedimentation, seasonal measurements were taken of surface elevation changes relative to the erosion needles.

## Soil physicochemical analyses

Collected soil samples were air-dried, gently crushed, and sieved (<2 mm) prior to laboratory analysis. Subsamples were further ground to <0.15 mm for carbon analyses. SIC (%): Measured using the gas volumetric method. Soil samples were reacted with hydrochloric acid, releasing $CO_2$ from carbonate decomposition, and the volume of $CO_2$ was used to calculate SIC content. SOC (g kg$^{-1}$): Determined using an elemental analyzer (dry combustion method). Soil samples were combusted at high temperature, and the released $CO_2$ was quantified to calculate total carbon (TC). SOC content was obtained by subtracting SIC from TC. Soil water content (WC, %): Measured in situ using a time domain reflectometry (TDR) moisture analyzer. Soil bulk density (BD, g cm$^{-3}$): Determined using the core method, by oven-drying intact soil cores at 105 °C for 24 h. Soil particle size distribution: Analyzed by the laser diffraction method after removal of organic matter and carbonates, including sand (2–0.05 mm, %), silt (0.05–0.002 mm, %), and clay (<0.002 mm, %). Soil pH: Measured in a 1:2.5 soil to water suspension using a glass electrode pH meter. Soil Compactness (penetration resistance, kPa): Determined in situ using a SC-900 (Spectrum Technologies Inc., USA) handheld soil compactness meter.

## Statistical analysis

First, Wilcoxon non-parametric tests were conducted to determine differences in soil properties, SOC, SIC, and TC stocks among different land use types, seasons, and depths. Second, the effects of land use types, seasons, depths, and their interactions on SOC, SIC, and TC stocks were tested using repeated measures analysis of variance (RMANOVA). Third, Pearson correlation analysis was used to assess the relationships between soil properties. Mantel tests were performed to evaluate the relationships between SOC, SIC, and TC stocks with soil properties for different depths. Finally, random forest modeling was used to evaluate the predictive power of environmental factors for SOC, SIC, and TC stocks among different land use types. Model cross-validation was performed using 70% of the data set for training and the remaining part for validation at each resampling iteration. We used 500 trees for model training (ntree value) and 3 predictors sampled at each node for splitting (mtry value), chosen to minimize RMSE [35]. Statistical significance was established at $p < 0.05$. All data processing, statistical analyses, and visualisation were conducted in R (version 4.4.2, R Foundation for Statistical Computing, Vienna, Austria), using packages *tidyverse, ggsignif, linkET, vegan, plyr, randomForest*, and *ggplot2*.

## Results

### Differences of soil properties under different land use types and seasons

Soil properties exhibited significant differences under different land use types and seasons (Table 1). Soil bulk density (BD) ranged from 1.40 to 1.64 g cm$^{-3}$ across all treatments, with cropland showing relatively lower BD (1.40 ± 0.30 g cm$^{-3}$), while other land use types maintained higher BD. Soil pH values were consistently alkaline, ranging between 7.7 and 8.1, cropland was significantly higher than shrubland ($p < 0.05$). The wind erosion index (WEI) varied substantially among land use types and seasons, with the highest values observed in grassland10 (23.14–34.24) and cropland (25.21–49.21), while shrubland consistently maintained the lowest values (6.00–18.88). Except for grassland10, the values in spring or winter were significantly higher than those in summer or autumn across all other land use types ($p < 0.05$). Plant cover varied significantly by season and land use. Grassland20 maintained the highest cover ($p < 0.05$), especially in summer (77.69 ± 8.33%) and spring (68.13 ± 6.05%), whereas cropland consistently exhibited the lowest cover ($p < 0.05$), with

no vegetation recorded in spring and winter due to cultivation practices. Differences in erosion thickness were mainly observed among land use types ($p < 0.05$), showing a trend of cropland > grassland10 > grassland20 > shrubland, but with no significant seasonal variation ($p > 0.05$). Soil water content (WC) and compactness also showed considerable variation but showed no spatiotemporal patterns, except that WC was significantly higher in winter due to snowfall (5.34%–8.11%). Soil texture showed relatively stable patterns. Sand content dominated soil texture across all treatments, with cropland had the lowest sand content (75.1%–83.6%), while other land use type consistently >93%. Clay content ranged between 1.2%–10.0%, with cropland generally higher than other land uses. Silt content varied between 2.7%–14.9%, with cropland also recording higher proportions.

## Differences in SOC, SIC, and TC stocks among different land use types

SOC, SIC, and TC exhibited significant variations among different land use types, and these differences were further influenced by season (Fig 2). SOC content varied significantly among land use types, and the differences were particularly amplified in autumn. In autumn, SOC in cropland, grassland10, grassland20, and shrubland were $3.0 \pm 0.3$, $4.0 \pm 1.2$, $4.4 \pm 1.5$, and $2.4 \pm 0.8$ g kg$^{-1}$, respectively, with significant differences among all land uses except between grassland10 and grassland20 ($p < 0.05$). In other seasons, SOC content in shrubland was the lowest (2.4–2.5 g kg$^{-1}$). Specifically, shrubland SOC was significantly lower than other land use types in winter, lower than cropland and grassland20 in spring, and only lower than grassland20 in summer. For SIC, cropland maintained consistently higher SIC content across all seasons, ranging from 11.6% to 12.4%, which was significantly greater than grassland10 (2.8%–2.9%), grassland20 (2.5%–3.1%), and shrubland (2.4%–2.6%) ($p < 0.05$). TC in the 1–5 cm soil layer also varied significantly with land use type. Across all seasons, cropland maintained the highest TC (4.2–4.7 kg C m$^{-2}$), significantly greater than other land use types ($p < 0.05$), but among other land use types were no significant differences.

Soil depth modulated the distribution of SOC, SIC, and TC among land use types (Fig 3). For cropland, SOC and SIC contents remained relatively similar between depths, fluctuating within 3.1–3.2 g kg$^{-1}$ and 11.6%–12.6%, respectively, and no significant differences ($p > 0.05$). And among other land use types, SOC showed consistently higher contents at the 0–1 cm than at the 1–5 cm, whereas the opposite pattern was observed for SIC and TC. At the 0–1 cm, except for cropland and grassland10, significant differences of SOC were observed among all land uses ($p < 0.05$). The SIC content of cropland was consistently significantly higher than that of other land use types throughout the 0–5 cm soil layer ($p < 0.05$). In addition, the SOC content of shrubland was significantly lower than that of grassland in the 0–1 cm soil layer ($p < 0.05$). TC is controlled by soil depth, so TC within 1–5 cm is always higher than TC within 0–1 cm. At the 1–5 cm, TC were $7.2 \pm 1.3$, $2.2 \pm 0.4$, $2.0 \pm 0.4$, and $1.9 \pm 0.7$ kg C m$^{-2}$ in cropland, grassland10, grassland20, and shrubland, respectively. And at the 0–1 cm, TC stocks were $1.7 \pm 0.4$, $0.5 \pm 0.1$, $0.5 \pm 0.1$, and $0.4 \pm 0.1$ kg C m$^{-2}$ in cropland, grassland10, grassland20, and shrubland, respectively.

## Driving mechanisms of SOC, SIC, and TC stock variability

Analysis of variance partitioning revealed that soil depth, land use, and their interactions were the dominant factors regulating SOC and TC variability (Fig 4a, 4c). And land use was the main factor regulating SIC variability (Fig 4b). For SOC, soil depth (60.86%), land use (27.54%), and their interaction (7.93%) together explained over 90% of the observed variation. Similarly, TC variation was mainly attributable to soil depth (64.97%), land use (25.04%), and their interaction (9.72%), accounting for nearly all variability. In contrast, SIC was overwhelmingly controlled by land use, which contributed 98.7% of the total explained variance, whereas soil depth and season interactions played a negligible role.

The correlations between SOC, SIC, TC and soil properties across the whole 0–5 cm layer and at specific depths revealed both consistent patterns and clear depth-dependent differences (Figs 5 and S1 Fig). At the profile scale, SOC variation was positively associated with soil pH, WEI, erosion thickness, clay, and silt, but negatively with sand. SIC variation showed much stronger associations, being positively correlated with pH, erosion thickness, clay, and silt, while

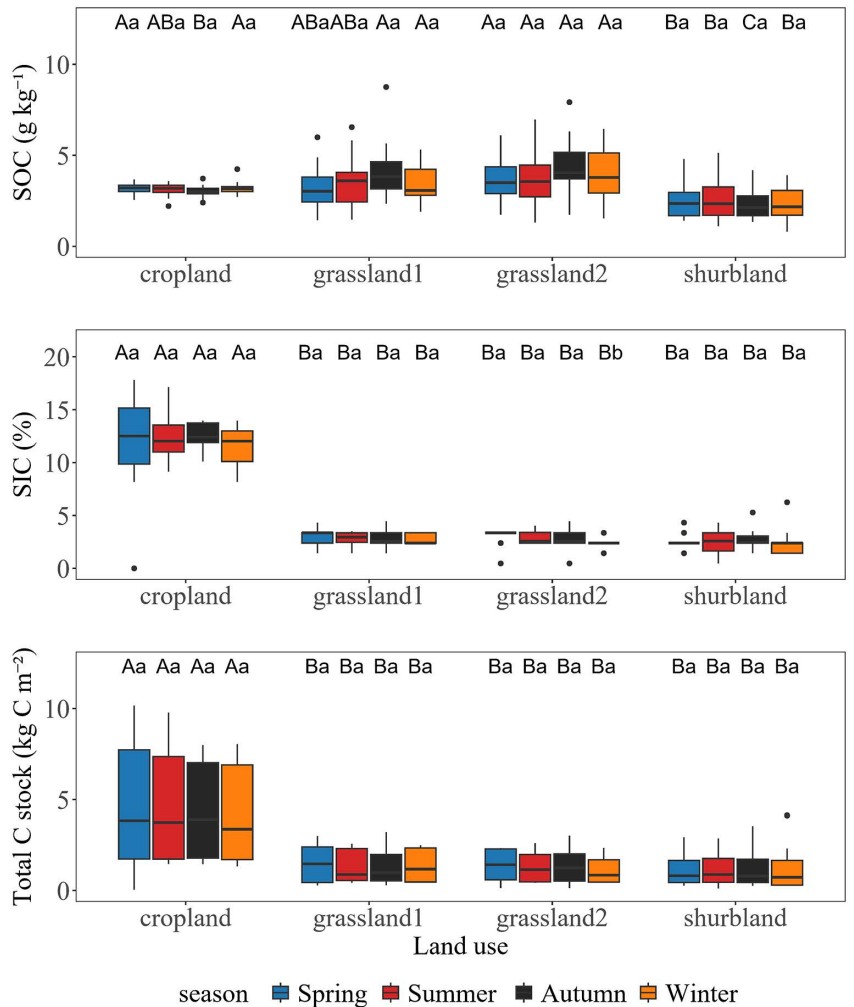

**Fig 2. The SOC, SIC, and TC stocks among different land use types vary in different seasons.** Different uppercase letters (A, B, C) within the same column indicate significant differences among land use types within the same seasons, while different lowercase letters (a, b, c) indicate significant differences among seasons within the same land use type ($p < 0.05$).

negatively correlated with BD, compactness, plant cover, and sand. TC largely mirrored SOC, being positively correlated with pH, WEI, erosion thickness, clay, and silt, but negatively correlated with BD, plant cover, compactness, and sand. At both 1 cm and 5 cm, SIC and TC demonstrated remarkably similar correlation structures, being positively related to pH, erosion thickness, clay and silt, and negatively related to BD, plant cover, compactness, and sand. WEI was also positively correlated with TC at both depths. However, SOC exhibited different patterns between layers. At 1 cm, SOC was significantly positively correlated with pH, WEI, erosion thickness, clay, and silt, but negatively correlated with BD and sand. In contrast, at 5 cm, SOC lost most of these associations and was only significantly positively correlated with plant cover. WC showed no significant correlation with any carbon component at either depth.

### Prediction and controlling factors of SOC, SIC, and TC stocks in wind erosion regions

The random forest model predicted SOC stocks with good accuracy, and the observed versus predicted values generally followed a linear relationship (Fig 6). Among land use types, shrubland showed the best fit ($R^2 = 0.38$), although the model

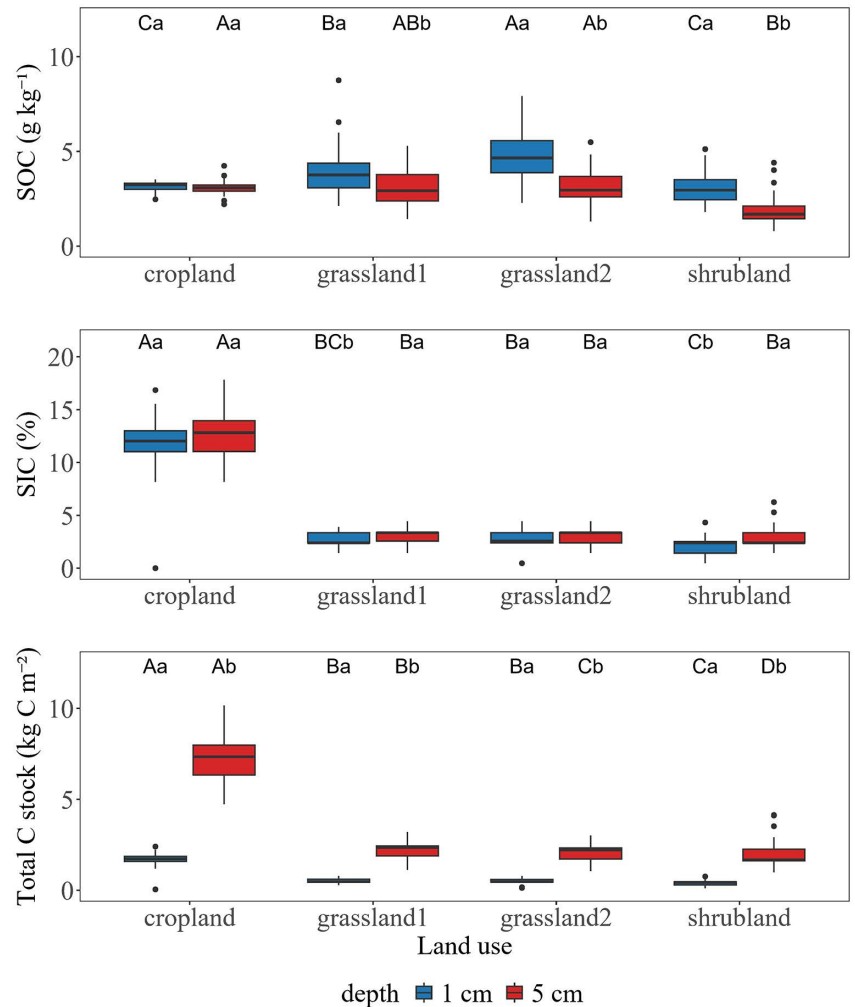

**Fig 3. The SOC, SIC, and TC stocks among different land use types vary in different depths.** Different uppercase letters (A, B, C) within the same column indicate significant differences among land use types within the same seasons, while different lowercase letters (a, b, c) indicate significant differences among seasons within the same land use type ($p < 0.05$).

tended to underestimate SOC at higher observed values. Cropland had the lowest $R^2$, but also the smallest prediction errors (RMSE = 0.27; MAE = 0.19). Variable importance analysis (increase in MSE) indicated that plant cover, soil compaction, and WC were the dominant predictors of SOC variability in cropland. In grassland20 and shrubland, soil texture (silt, sand, and clay) was the most influential factor, whereas in grassland10, prediction performance was relatively balanced across factors, with compaction exerting a slighter effect.

Model performance for SIC was generally weaker compared to SOC, but still acceptable for most land use types (Fig 7). Except for cropland, predictions of other land use types matched observations (RMSE < 0.8; MAE < 0.6). In particular, grassland10 showed a very low $R^2$ (0.04), indicating poor predictive ability. Across all land use types, soil texture (silt, sand, and clay) consistently emerged as the primary determinant of SIC variability, with additional contributions from water content in grassland20.

The random forest model performed poorly in predicting TC in cropland and grassland10 ($R^2 < 0.1$), but showed strong performance in grassland20 ($R^2 = 0.80$) (Fig 8). Soil texture (silt, clay, and sand) were consistently the most important predictors, highlighting the synergistic role of particle size composition in regulating TC variability.

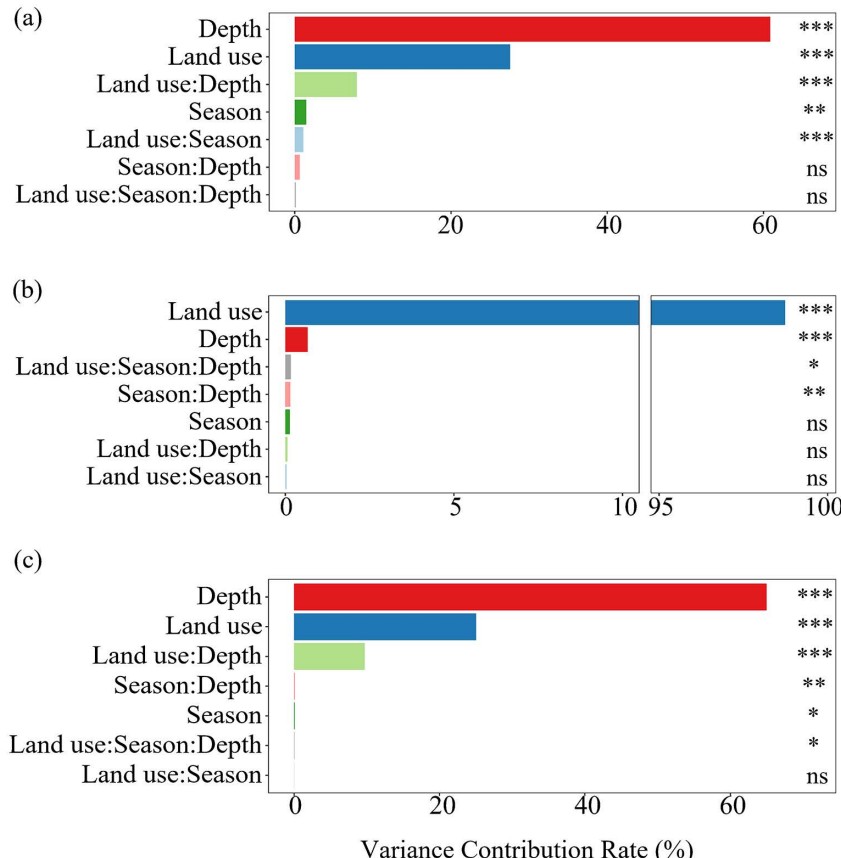

**Fig 4. Contribution of depths, land use types, and seasons on soil carbon variance.** (a) drivers of SOC variance, (b) drivers of SIC variance and (c) drivers of TC variance. *, **, *** means significant differences between layers at the 0.05, 0.01, 0.005 level, respectively. Non-significant pairs are indicated (ns; $p > 0.05$).

## Discussion

### Effects of land use type on SOC, SIC, and TC variability

Here, the differences in soil carbon stocks (SOC, SIC) were observed among the different land uses. These results demonstrated clearly that the management of land significantly impacts carbon budgets in arid and semi-arid regions [4,21,36]. Among the four land uses, cropland exhibited the highest TC contents at 5 cm, largely attributable to elevated SIC levels. Long-term fertilization and tillage contributed to higher levels of alkalinization in arid areas. Thus, soil pH was higher compared to other land use, which favors carbonate deposition. Higher levels of alkalinity favor also SIC accumulation [20,37,38]. Thus, TC dynamics in croplands was controlled by inorganic carbon cycle. These results suggested that cultivation is the main factor controlling the SIC enrichment in the wind erosion areas.

On the contrary, grasslands presented a higher SOC storage compared with croplands and shrublands. The higher plant cover in grasslands minimized evapotranspiration [39], and consequently the soil water availability and primary productivity in drylands. This fact, in addition to sheltered wind erosion and increased organic matter input, consequently enhancing SOC accumulation in the surface horizon. [19,40]. Consistent with studies from rangeland systems in the western US, the low-vegetated bare ground losses of SOC due to wind erosion are already substantially diminished through

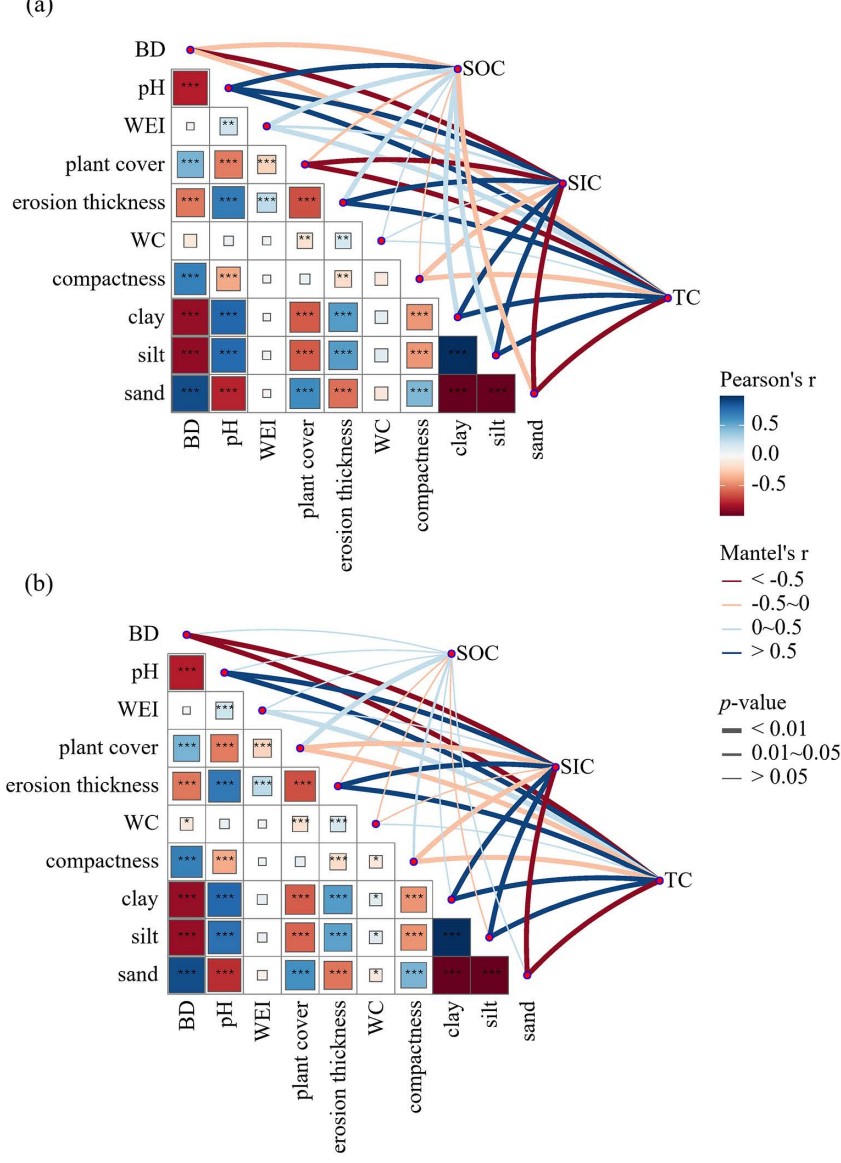

**Fig 5. The relationship between soil properties and SOC, SIC, and TC by Mantel tests.** (a) at 1 cm depth and (b) at 5 cm depth. BD, bulk density (g cm$^{-3}$); WEI, wind erosion index; WC, soil water content (%); SOC, soil organic carbon; SIC, soil inorganic carbon; TC, total carbon stock. The thickness of lines indicates the significance of Mantel tests, with thicker lines indicating more pronounced significance. The color of lines indicates the correlation coefficient of Mantel tests, while blue and red represent positive and negative correlations, respectively. *, **, *** means significant differences between layers at the 0.05, 0.01, 0.005 level, respectively.

relatively small vegetation cover increases [41]. These findings emphasized the key role of vegetation cover for maintaining SOC in this type of environment.

Another interesting finding is that, despite the relatively high canopy cover achieved under ecological restoration, shrublands exhibited the lowest SOC levels, likely due to the dominance of recalcitrant shrub litter and the slow turnover of aboveground biomass and root systems, which together limit organic matter inputs [31,42]. The temporal lagging of SOC accumulation in the early stage of shrub revegetation has also been found previously [43]. SOC accumulation was

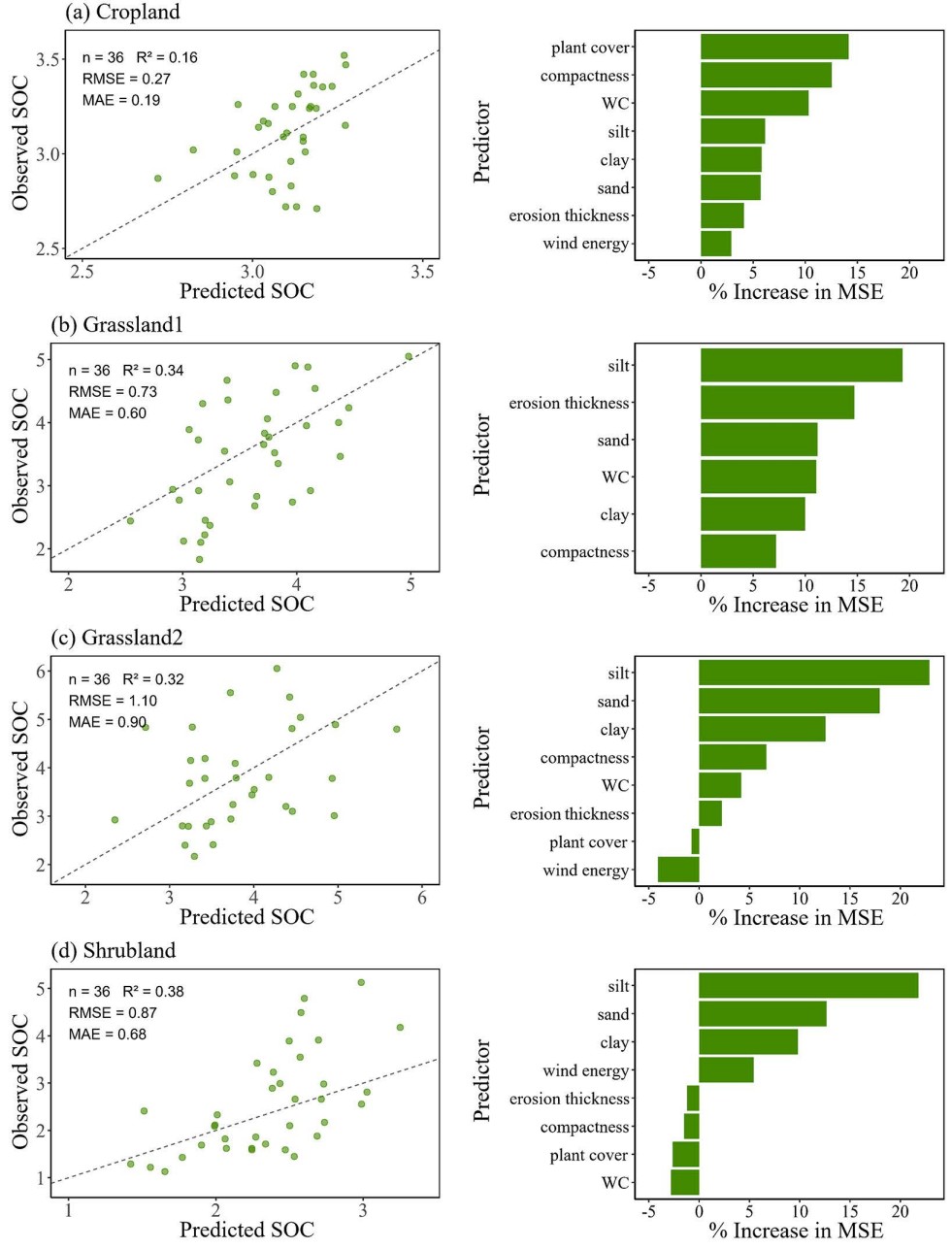

**Fig 6. SOC prediction and variable importance by random forest.** (a) cropland, (b) grassland10, (c) grassland20, and (d) shrubland.

only detectable after decades of vegetation succession, indicating that establishment of shrubs with rapid succession might not bring about immediate SOC accumulation, whereas grassland would potentially recover SOC faster.

In summary, our results underlined the different land use effects on controlling soil carbon pools. Crop cultivation contributed to increase SIC storage due to increasing alkalinity, grassland restoration was contributing to increase SOC storage and improving ecosystem resistance to wind erosion, while the establishment of shrubs could initially hinder the SOC storage because of low litter decomposition rate. These different paths have proved that soil carbon responses in arid areas at different land use types were mainly regulated and had important impact on long term carbon sequestration strategies.

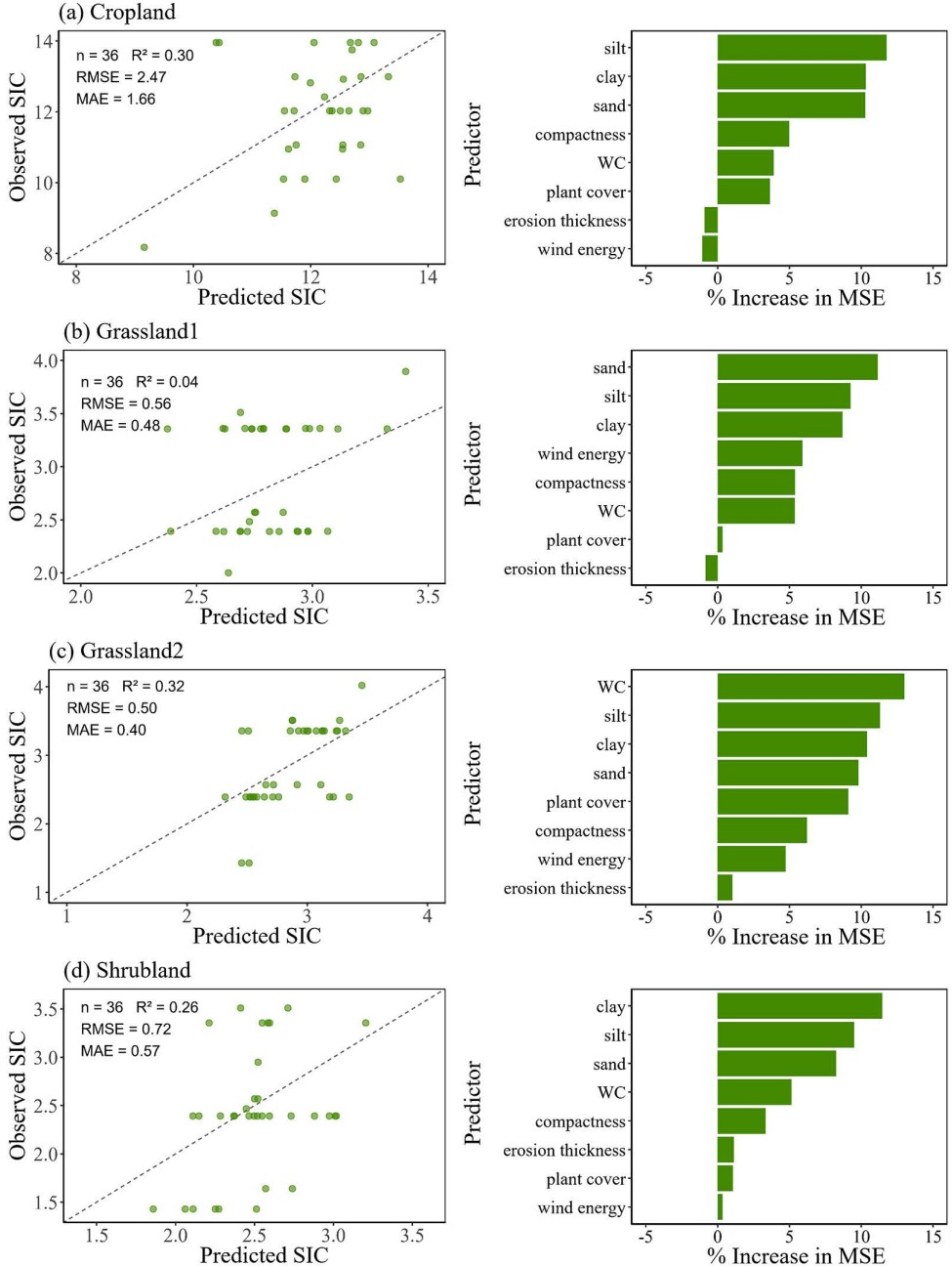

**Fig 7. SIC prediction and variable importance by random forest.** (a) cropland, (b) grassland10, (c) grassland20, and (d) shrubland.

## Depth variability and wind erosion effect

Depth emerged as a significant determinant of SOC and SIC variability (Fig 4). This finding was consistent with previous evidence that surface soils were the most sensitive to both aeolian disturbance and land management [13,41]. SOC was consistently enriched at the 1 cm, reflecting the dominant role of litter inputs, root exudation, and microbial turnover in maintaining surface SOC pools [44]. In contrast, SIC and TC were relatively concentrated at the 5 cm, indicating downward leaching of carbonates and the preferential redistribution of fine, carbonate-rich particles under wind erosion [16,45].

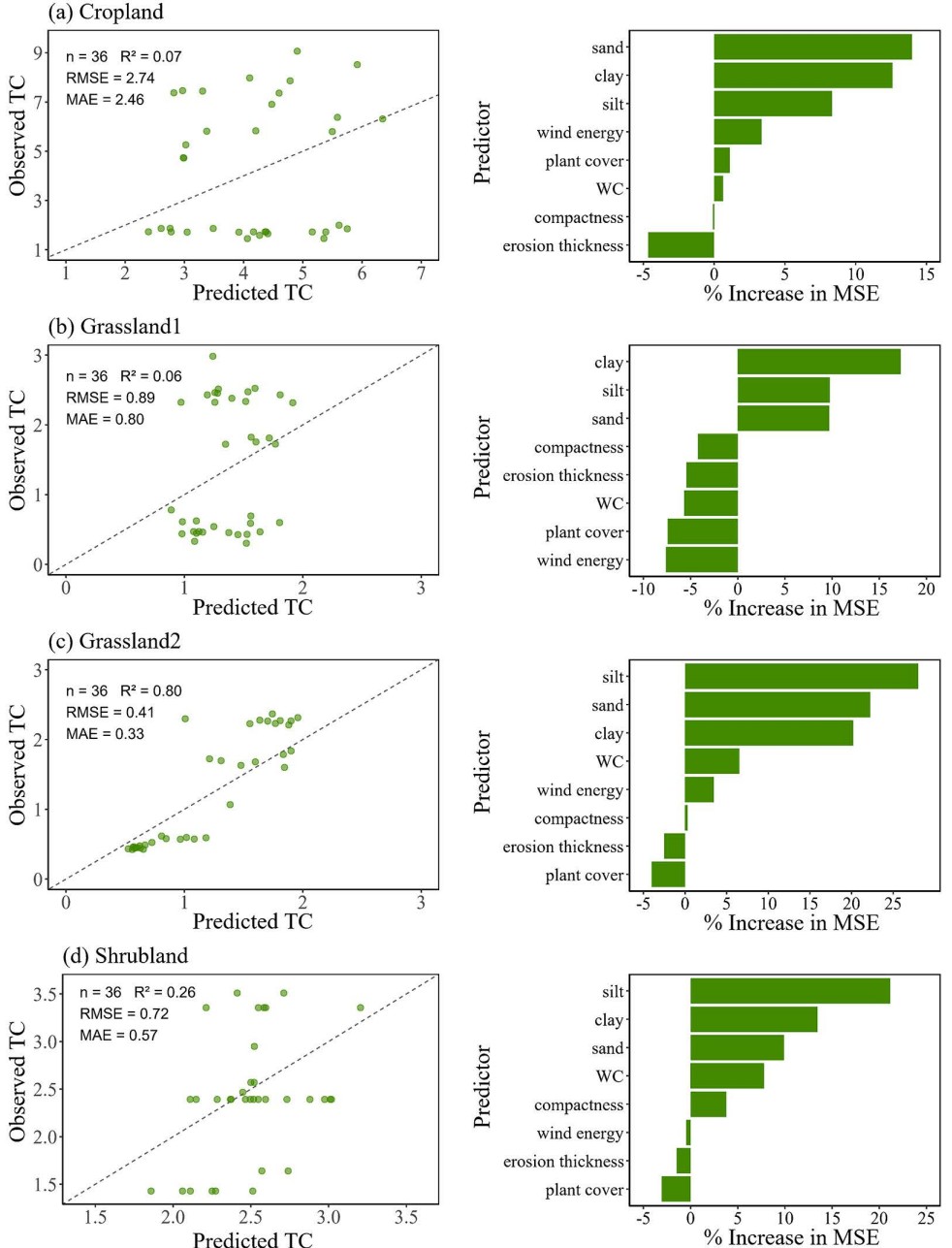

**Fig 8. TC prediction and variable importance by random forest.** (a) cropland, (b) grassland10, (c) grassland20, and (d) shrubland.

Such contrasting depth profiles highlighted the divergent stabilization mechanisms of SOC and SIC in wind erosion landscapes.

Wind erosion further amplified SOC differences among land uses in the surface layer [16,17]. For example, SOC contents in the minimally eroded grassland20 were significantly higher than in the more eroded grassland10 and shrubland at 1 cm. This pattern was consistent with previous observations from temperate grassland, where aeolian processes selectively removed fine, carbon-rich particles from the surface, leading to SOC depletion and greater heterogeneity across

land uses [13]. In croplands, however, such vertical contrasts were largely absent, which can be explained by tillage induced soil mixing that disrupts natural stratification [21,36].

For SIC, depth related differences were also evident, though less pronounced than for SOC. Across all land uses, SIC contents at 1 cm were consistently lower than at 5 cm, in line with the typical leaching driven accumulation of carbonates in subsurface layers [46,47]. Significant SIC differences between 1 and 5 cm were only detected in grassland10 and shrubland, but not in grassland20, suggesting that stronger wind erosion in the former two land uses reshaped vertical SIC distribution. Notably, in grassland20, SIC exhibited marked seasonal variability, being significantly lower in winter. This can be attributed to reduced plant cover and higher wind erosion intensity during winter, which intensified carbonate redistribution [16,17]. In shrublands, despite their relatively dense canopy, SIC depth contrasts were also significant, likely reflecting vegetation induced changes in soil hydrology. Previous studies have noted that shrubs with high water demand can exacerbate erosion by depleting soil moisture and weakening protective biocrusts [48,49].

Taken together, these results demonstrated that wind erosion exerted its greatest influence on surface soils, attenuating the vertical stratification that would otherwise arise from litter accumulation and carbonate leaching. Land use conversion from cropland to grassland was shown to mitigate these erosional effects, favoring the accumulation of both SOC and SIC. By contrast, shrub encroachment did not enhance SOC or SIC in the short term and may even intensify erosion risk under water limited conditions. These findings underscored the importance of maintaining continuous herbaceous cover as a strategy to stabilize both SOC and SIC pools in arid wind erosion regions.

## Influence factors of SOC, SIC in wind erosion regions

The main regions identified as prone to wind erosion also corresponded with those with limited vegetation cover and fragile soils, whose destruction and alteration would further increase the vulnerability to wind erosion processes, thus related carbon loses [13,18,50]. In terms of soil chemical attributes, pH resulted a better predictor of SIC dynamics (Fig 5). The positive association between soil pH and SIC found in this study corroborates previous evidence that carbonate formation and accumulation are promoted under alkaline soil conditions [30,37]. Soil carbon stocks dominated by $HCO_3^-$ and $CO_3^{2-}$ with soil pH higher than 7 would be driven to precipitate SIC [12]. Also, soil pH was buffered by carbonates in soil alkaline systems; calcium carbonate could neutralize hydrogen ions in alkaline environment, stabilizing soil pH and preventing soil further acidification [11]. Lower soil pH diminishes the rate of carbonate dissolution, thereby limiting the formation and storage of SIC. However, this was potentially limited in arid systems by insufficiency of water. Under sufficient soil moisture, $CO_2$ forms $HCO_3^-$, which can combine with $Ca^{2+}$ to generate SIC [51]. Therefore, under adequate soil moisture, the equilibrium of dissolution and precipitation of SIC in drylands was controlled by pH and water content.

Soil texture also played a key role. We have observed a negative effect of soil with a high sand content on SOC, SIC and TC, whereas the increase of clay and silt contents had positive effects on them (Fig 5b). This was consistent with the idea that sandy soils due to their greater porosity and permeability were not good in terms of carbon holding ability [52]. Reduced clay and silt contents tended not only to decrease water-holding capacity, but also to decrease the amount of $Ca^{2+}$ adsorbed thereby limiting the potential to both SOC stabilization and SIC precipitation [8]. Random forest predictions also suggested that texture was again the most important predictor of variation in SOC, SIC and TC across different land use types with clay and silt identified as the most important predictors (Figs 6–8). Explanation power of texture was not significant for prediction except for SOC in croplands. Texture had a lower level of contribution than expected probably because intense plowing homogenized particle distribution, decreasing explanatory power of soil texture. The case of croplands suggested that even though soil texture was an overall control on the quantity of carbon pools in areas exposed to wind erosion, anthropic behavior could influence or even supersede its importance [53].

Depth dependent differences in controlling factors were also evident. At 1 cm, SOC and SIC were strongly correlated with wind erosion indicators (WEI and erosion thickness), confirming the high sensitivity of surface soils to erosional processes and selective particle redistribution [41]. At 5 cm, however, SOC was no longer associated with erosion

indicators but was mainly governed by vegetation cover, suggesting that subsurface SOC stabilization relies on sustained plant inputs [8]. Interestingly, vegetation cover was negatively correlated with SIC at both 1 cm and 5 cm, which may be explained by plant secretion of organic acids that convert SIC into $CO_2$ under alkaline conditions [54]. This indicated a potential trade off in which vegetation restoration enhanced SOC sequestration but can reduced SIC storage in calcareous soils.

Overall, the dynamics of soil carbon in this wind-erosion-prone landscape emerged from complex interactions among biotic, abiotic, and anthropogenic drivers. The 0–1 cm layer was highly sensitive to erosion-driven redistribution, whereas the 1–5 cm layer depended more strongly on continuous plant-derived inputs. In contrast, SIC dynamics were strongly governed by soil pH and texture, where alkaline conditions promoted carbonate precipitation and finer particles (clay and silt) enhanced $Ca^{2+}$ retention and SIC formation. Soil moisture availability further modulated SIC dissolution-precipitation, particularly under limited rainfall conditions. Soil texture acted as a master variable influencing both SOC stabilization and SIC accumulation, though its effect could be moderated by land use such as cropping and tillage. Vegetation restoration exhibited a dual role that it enhanced SOC sequestration by reducing erosion and adding organic matter, yet also risked lowering SIC through root-induced acidification. From a management perspective, this highlighted the need for integrated strategies that simultaneously stabilize SOC and SIC. Promoting continuous herbaceous cover can effectively reduce wind erosion and enhance SOC accumulation, whereas controlling alkalization and maintaining sufficient soil moisture were essential for moderating SIC dynamics. Sustainable land use practices in wind-eroded regions must account for the tightly coupled responses of both carbon pools to erosion, vegetation, and soil properties, recognizing their nonlinear interactions and long-term feedbacks.

## Uncertainty and limitation

Despite this paper being the first to reveal spatiotemporal dynamics of SOC and SIC of different types of land uses in wind erosion areas, there are the limitations that we should recognize. First, high uncertainty of predictive power from the random forest model for obtaining the importance of different land use types was significant, for which the accurate predictions based on land use are difficult to follow. Specifically, explanatory power for SOC was low in croplands suggesting that the model was limited by unmeasured factors such as microbial processes, carbonate dissolution like fertilized. Additionally random forests are susceptible to overfitting, which necessitates combining process-based models with statistical prediction. Second, the monitoring period was only two years, which might not be sufficient to account for interannual variability in the wind regimes and vegetation evolution and extreme events such as dust storms or dry years. Consequently, representativeness of our results for longer time scale is not clear. Long term observational networks and chrono sequence methods would be helpful to decipher the short-term variation from long term trend. Third, we limited our sampling to the 0–5 cm horizon because this topsoil is directly impacted by wind erosion. Although our aim was to clarify the surface processes in operation here, the results may underestimate contributions of the deeper horizons that often contain large SIC reserves and stable SOC pools. By not sampling through the profile, our estimates of TC stocks and particularly our observations of long-term stabilization processes remain uncertain [15]. Addressing those uncertainties in future work includes further extending both temporal and spatial coverage. Longer monitoring periods across more climatic conditions will strengthen estimation of carbon fluxes. Deeper soil sampling would include subsurface soil carbon pools into regional budgets. Technically, coupling of isotopic tracers with laboratory experiments under a control condition will aid in distinguishing the addition of new or old carbon and will assist in the separation between biotic and abiotic stabilization processes. Coupling field observations with wind tunnel simulation as well as process-based erosion and carbon models should also enable us a more mechanistic picture of SOC and SIC redistribution under varying land use and climate regimes. Such limitations of the studies would be recognized and addressed by multi scale and interdisciplinary studies in the future, leading to better assessments of the fate of soil carbon pools in dryland ecosystems, and a stronger basis for scientific advice on land management and C sequestration strategies. Even with such limitations in our study,

there are still multiple new insights we obtained. We show that the risk of erosion can be minimized and soil carbon pools can be stabilized by ecological restoration, which provides convincing evidence that land use managements is an effective measure to mitigate the land degradation and promote regional carbon neutrality.

## Conclusion

This study systematically examined the spatiotemporal dynamics and controlling factors of SOC, SIC, and TC under different land use types in a typical wind-eroded region of northern China. By multi-seasonal field observations, we found that land use change profoundly altered soil carbon pools. Grassland restoration was more effective than shrub restoration in wind-eroded regions for enhancing SOC storage. The recovery strategy should comprehensively consider the impact of SOC and SIC, integrate them into the regional carbon budget. The surface layer is most susceptible to wind erosion, and protective measures such as stubble retention and checkerboard barriers should be strengthened to maintain surface carbon stock. In summary, these findings advance our understanding of the influence mechanisms regulating SOC and SIC in wind-eroded regions. SOC is largely controlled by biotic processes and erosion dynamics. SIC is primarily governed by geochemical buffering. And soil texture modulates both them. Importantly, surface soils as the critical interface where vegetation, erosion, and soil chemistry interact, shaping the short-term variability and long-term result of soil carbon.

## Supporting information

**S1 Fig. The relationship between soil properties and SOC, SIC, and TC by Mantel tests.** BD, bulk density (g cm$^{-3}$); WEI, wind erosion index; WC, soil water content (%); SOC, soil organic carbon; SIC, soil inorganic carbon; TC, total carbon stock. The thickness of lines indicates the significance of Mantel tests, with thicker lines indicating more pronounced significance. The color of lines indicates the correlation coefficient of Mantel tests, while blue and red represent positive and negative correlations, respectively. *, **, *** means significant differences between layers at the 0.05, 0.01, 0.005 level, respectively.
(TIF)

**S1 File. Raw images.**
(ZIP)

## Acknowledgments

We would like to express our sincere gratitude to the local farmers for their invaluable support and cooperation throughout the field study. We are especially grateful for their permission to access the study sites and for their voluntary assistance in the routine maintenance of the experimental plots.

## Author contributions

**Conceptualization:** Bin Xia, Wei Xu.

**Data curation:** Bin Xia.

**Formal analysis:** Bin Xia.

**Funding acquisition:** Wei Xu.

**Investigation:** Bin Xia, Wei Xu.

**Methodology:** Bin Xia.

**Project administration:** Wei Xu.

**Resources:** Bin Xia.

**Software:** Bin Xia.

**Supervision:** Bin Xia, Wei Xu.

**Validation:** Bin Xia, Wei Xu.

**Visualization:** Bin Xia.

**Writing – original draft:** Bin Xia.

**Writing – review & editing:** Bin Xia, Wei Xu.

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
