## [Decision Letter · Decision Letter 0]

19 Feb 2026

Dear Dr. Xia,

Thank you for submitting your manuscript to PLOS ONE. After careful consideration, we feel that it has merit but does not fully meet PLOS ONE’s publication criteria as it currently stands. Therefore, we invite you to submit a revised version of the manuscript that addresses the points raised during the review process.

We look forward to receiving your revised manuscript.

Kind regards,

Marcela Pagano, Ph.D, M.D.

Academic Editor

PLOS One

Journal Requirements:

https://journals.plos.org/plosone/s/file?id=wjVg/PLOSOne_formatting_sample_main_body.pdf and and and and https://journals.plos.org/plosone/s/file?id=ba62/PLOSOne_formatting_sample_title_authors_affiliations.pdf

5. We note that Figure 1 in your submission contain copyrighted images. All PLOS content is published under the Creative Commons Attribution License (CC BY 4.0), which means that the manuscript, images, and Supporting Information files will be freely available online, and any third party is permitted to access, download, copy, distribute, and use these materials in any way, even commercially, with proper attribution. For more information, see our copyright guidelines: http://journals.plos.org/plosone/s/licenses-and-copyright.

6. We notice that your supplementary [figure 1] are included in the manuscript file. Please remove them and upload them with the file type 'Supporting Information'. Please ensure that each Supporting Information file has a legend listed in the manuscript after the references list.

Reviewers' comments:

Reviewer's Responses to Questions

**Comments to the Author**

1. Is the manuscript technically sound, and do the data support the conclusions?

Reviewer #1: Yes

2. Has the statistical analysis been performed appropriately and rigorously?

Reviewer #1: Yes

3. Have the authors made all data underlying the findings in their manuscript fully available?

Reviewer #1: Yes

4. Is the manuscript presented in an intelligible fashion and written in standard English?

Reviewer #1: Yes

Reviewer #1: The manuscript is well written and results are justified. Kindly refer to the manuscript for minor comments.

Abstract is well written,

1. Also highlight other findings parameters observed than SOC & SIC

2. The implications of these results?

Introduction nicely covered the study background, importance and hypothesis. 1. However the review of literature in the field needs to be elaborated

2. Objectives of the study are missing.

.

Reviewer #1: No

---

## [Author Response · Author response to Decision Letter 1]

3 Mar 2026

Response to Reviewer:

Comments 1: Introduction: Line 57: a-1 ??

Responses: We appreciate your careful attention to detail. To ensure consistency and avoid confusion, we have now revised the manuscript throughout (line 76): replacing instances of a-1 with yr-1. This modification aligns with the standard notation used in subsequent sections.

Comments 2: Introduction: Line 71: Grasslands abandoned for 20 years, but how its being used presently?

Responses: We agree that the description of the land use types could be clarified. The grasslands abandoned for 10 and 20 years were previously cultivated croplands that were taken out of agricultural production and subsequently left for natural restoration. Currently, these sites are managed as permanent grasslands without any use such as grazing or mowing. The shrubland was established by planting shrubs on former cropland as part of ecological restoration efforts. We have revised the sentence accordingly to avoid ambiguity (Line 89-93): cropland (continuously cultivated), grasslands abandoned for 10 and 20 years (naturally restored after cultivation cessation), and shrubland (established through afforestation on former cropland). Soil samples were collected at two key surface depths (1 cm and 5 cm) across multiple seasons.

Comments 3: Introduction: Line 81-84: introduction nicely covered the study background, importance and hypothesis. 1. However the review of literature in the field needs to be elaborated

2. Objectives of the study are missing.

Responses: 1. We sincerely thank the reviewer for the constructive suggestion. In response, we have substantially expanded the literature review to strengthen the theoretical background and improve the integration of our study within the broader field of soil carbon and wind erosion research. Specifically, the following revisions have been made:

Strengthening the global context of SOC and SIC (Lines 45–51): The SOC stocks to 2 m of soil depth are estimated at approximately 2400 Pg [5]. In arid and semi-arid regions, inorganic carbon stored as pedogenic carbonates can account for a substantial proportion of total soil carbon. soils store 2305 ± 636 Pg of carbon as SIC within the upper 2 m, a magnitude comparable to or even exceeding global SOC stocks. Importantly, this pool is not static. Projections suggest that global SIC stocks (0–30 cm) will reduce by up to 23 Pg of carbon [6; 7]. These additions strengthen the scientific justification for jointly examining SOC and SIC dynamics.

Enhancing the mechanistic background on wind erosion and surface soil vulnerability (Lines 66–68): Furthermore, small variations in the uppermost millimeters to centimeters of soil aggregation, crusting, and moisture can substantially alter erosion thresholds, thereby changing the redistribution process [22]. This addition reinforces the rationale for focusing on the 1 cm depth as a critical interface between wind energy and soil carbon pools.

Overall, these revisions provide a more comprehensive and quantitatively supported literature foundation, particularly by clarifying the global significance and underrepresentation of SIC research, and strengthening the theoretical linkage between wind erosion processes and surface carbon dynamics. We believe these improvements address the reviewer’s concern and substantially enhance the scholarly depth of the Introduction.

Reference:

D. Beillouin, M. Corbeels, J. Demenois, D. Berre, A. Boyer, A. Fallot, F. Feder, and R. Cardinael, A global meta-analysis of soil organic carbon in the Anthropocene. Nature Communications 14 (2023) 3700.

Y. Huang, X. Song, Y.-P. Wang, J.G. Canadell, Y. Luo, P. Ciais, A. Chen, S. Hong, Y. Wang, F. Tao, W. Li, Y. Xu, R. Mirzaeitalarposhti, H. Elbasiouny, I. Savin, D. Shchepashchenko, R.A.V. Rossel, D.S. Goll, J. Chang, B.Z. Houlton, H. Wu, F. Yang, X. Feng, Y. Chen, Y. Liu, S. Niu, and G.-L. Zhang, Size, distribution, and vulnerability of the global soil inorganic carbon. Science 384 (2024) 233-239.

S. Raza, A. Irshad, A. Margenot, K. Zamanian, N. Li, S. Ullah, K. Mehmood, M. Ajmal Khan, N. Siddique, J. Zhou, S.J. Mooney, I. Kurganova, X. Zhao, and Y. Kuzyakov, Inorganic carbon is overlooked in global soil carbon research: A bibliometric analysis. Geoderma 443 (2024) 116831.

Y. Gu, Y. Liu, P. Shi, G. Zhang, Y. Yang, G. Wang, Z. Hu, and L. Liu, Synchronous field measurement of high energy sand saltation on typical desert surfaces, Alxa plateau. Scientific Reports 15 (2025) 23302.

2. Thanks for your positive and constructive feedback. We agree that the objectives of the study should be explicitly stated. Following your suggestion, we have now added research objectives after the hypotheses (Line 102-107): This study was designed to quantify the spatiotemporal variations in SOC and SIC across different land use types, soil depths, and seasons, to compare their distribution patterns between the surface soil layer (0–1 cm) most directly affected by wind erosion and the subsurface layer (1–5 cm), and to disentangle the relative contributions of wind erosion versus land use practices in shaping SOC and SIC heterogeneity while elucidating the underlying mechanisms. These objectives are directly aligned with the hypotheses and outline the specific steps we took to address them.

Comments 4: Materials and methods: Line 97: Can you explain gale days ?, because its not common to all.

Responses: We have now clarified definition of gale days in the Materials and methods (line 123): days with daily maximum wind speed ≥17.0 m s-1 at the 10 m height [28] and added the reference. This revision improves the clarity of the manuscript for an international audience.

Reference:

R. Yuan, Q. Li, L. Wu, M. Huo, and Y. Huang, Evaluation and Projection of Gale Events in North China, Atmosphere, 2023, pp. 1646.

Comments 5: Materials and methods: Line 168-176: Overall, the M & M section is complete, but management practices under different land management systems over the years (from the beginning) is missing. Since these practices governs the erodability and soil carbon fractions.

Responses: Thanks for your comment regarding the missing historical management practices. In response, we have expanded the Methods section to include detailed descriptions of the land-use history and current management practices for each of the four land use types in the Materials and methods (Line 139-149): 1) cropland continuously cultivated under traditional farming, and dominated by Fagopyrum esculentum Moench. 2) grassland10 abandoned from cropland in 2011 (10 years prior to sampling), and dominated by Artemisia scoparia, Leymus secalinus, Heteropappus altaicus. 3) grassland20 abandoned from cropland in 2001 (20 years prior to sampling) and dominated by Agropyron cristatum, Leymus secalinus, Heteropappus altaicus. Both grassland10 and grassland20 have naturally recovered without any human interference (e.g., grazing, mowing, or reseeding) since abandonment. Before abandonment, they were managed identically to the cropland. 4) shrubland converted from cropland in 2011 (10 years prior to sampling) by planting shrubs (dominated by Salix pasmmophara, Hedysarum scoparium), it has been protected from grazing and other disturbances since establishment. Specifically, we now describe the continuous cultivation history of the cropland, the identical pre-abandonment cultivation of the two grasslands and shrubland, and the post-abandonment management (natural restoration without anthropogenic disturbance for grasslands; shrub planting followed by protection from grazing for shrubland). We believe this addition provides the necessary context for interpreting the effects of land use and wind erosion on soil carbon pools. Thank you for helping us improve the clarity and completeness of our manuscript.

Comments 6: Result: Line 206-207: Mention about these rankings as a footnote.

Also give rankings as a lower case alphabets "a / ab / bc" instead of 'A / Ab".

This can be followed in all the tables.

Responses: Thanks for your comment regarding the significance rankings in the tables. We appreciate your suggestion to use lowercase letters uniformly. However, we would like to clarify that the uppercase and lowercase letters in our tables serve distinct purposes to efficiently convey two levels of comparison within a single table:

Uppercase letters (A, B, C) indicate significant differences among land use types within same seasons.

Lowercase letters (a, b, c) indicate significant differences among seasons within the same land use type.

This dual-letter system allows readers to quickly assess both cross-land-use and cross-seasonal variations without needing separate tables or additional columns. We have revised the footnote in line 241-242, line 277-279, line 282-284: Different uppercase letters (A, B, C) within the same column indicate significant differences among land use types within the same seasons, while different lowercase letters (a, b, c) indicate significant differences among seasons within the same land use type.

Comments 7: Discussion: Line 328: Results demonstrated.

Responses: Many thanks for the reviewer's suggestion and we agree that a more objective phrasing is appropriate in the discussion. In addition to accepting your revision in line 362: These results demonstrated clearly that the management of land significantly impacts carbon budgets in arid and semi-arid regions, we have also revised the preceding sentence in line 361: Here, the differences in soil carbon stocks (SOC, SIC) were observed among the different land uses.

We greatly appreciate the dedication of the reviewers in improving this work and are honored to implement any additional improvements. This revision process is an inspiring academic journey, and we remain committed to advancing the understanding of soil carbon pools through rigorous scientific exploration.

Once again, thank you very much for your comments and suggestions.

Sincerely,

Bin Xia, Wei Xu

Corresponding author:

Wei Xu

---

## [Editor Report · Decision Letter 1]

23 Mar 2026

Grassland Restoration in Typical Wind-eroded Regions Effectively Increase Soil Organic Carbon

PONE-D-25-60997R1

Dear Dr. Xu,

We’re pleased to inform you that your manuscript has been judged scientifically suitable for publication and will be formally accepted for publication once it meets all outstanding technical requirements.

Kind regards,

Marcela Pagano, Ph.D, M.D.

Academic Editor

PLOS One
---

## [Editor Report · Acceptance letter]

PONE-D-25-60997R1

PLOS One

Dear Dr. Xu,

I'm pleased to inform you that your manuscript has been deemed suitable for publication in PLOS One. Congratulations! Your manuscript is now being handed over to our production team.

Kind regards,

on behalf of

Dr. Marcela Pagano

Academic Editor

PLOS One